# An Optically Induced Dielectrophoresis (ODEP)-Based Microfluidic System for the Isolation of High-Purity CD45^neg^/EpCAM^neg^ Cells from the Blood Samples of Cancer Patients—Demonstration and Initial Exploration of the Clinical Significance of These Cells

**DOI:** 10.3390/mi9110563

**Published:** 2018-10-31

**Authors:** Chia-Jung Liao, Chia-Hsun Hsieh, Tzu-Keng Chiu, Yu-Xian Zhu, Hung-Ming Wang, Feng-Chun Hung, Wen-Pin Chou, Min-Hsien Wu

**Affiliations:** 1Graduate Institute of Biomedical Engineering, Chang Gung University, Taoyuan City 33302, Taiwan; l329735@ms49.hinet.net (C.-J.L.); maple7530@gmail.com (Y.-X.Z.); fjiun@hotmail.com (F.-C.H.); d94522010@ntu.edu.tw (W.-P.C.); 2Division of Haematology/Oncology, Department of Internal Medicine, Chang Gung Memorial Hospital (Linko), Taoyuan City 33302, Taiwan; wisdom5000@gmail.com (C.-H.H.); whm526@adm.cgmh.org.tw (H.-M.W.); 3Department of Chemical and Materials Engineering, Chang Gung University, Taoyuan City 33302, Taiwan; b74225@hotmail.com; 4Department of Chemical Engineering, Ming Chi University of Technology, New Taipei City 24301, Taiwan

**Keywords:** microfluidic systems, optically induced dielectrophoresis (ODEP), cell isolation, circulating tumour cells (CTCs), cancer metastasis

## Abstract

Circulating tumour cells (CTCs) in blood circulation play an important role in cancer metastasis. CTCs are generally defined as the cells in circulating blood expressing the surface antigen EpCAM (epithelial cell adhesion molecule). Nevertheless, CTCs with a highly metastatic nature might undergo an epithelial-to-mesenchymal transition (EMT), after which their EpCAM expression is downregulated. In current CTC-related studies, however, these clinically important CTCs with high relevance to cancer metastasis could be missed due to the use of the conventional CTC isolation methodologies. To precisely explore the clinical significance of these cells (i.e., CD45^neg^/EpCAM^neg^ cells), the high-purity isolation of these cells from blood samples is required. To achieve this isolation, the integration of fluorescence microscopic imaging and optically induced dielectrophoresis (ODEP)-based cell manipulation in a microfluidic system was proposed. In this study, an ODEP microfluidic system was developed. The optimal ODEP operating conditions and the performance of live CD45^neg^/EpCAM^neg^ cell isolation were evaluated. The results demonstrated that the proposed system was capable of isolating live CD45^neg^/EpCAM^neg^ cells with a purity as high as 100%, which is greater than the purity attainable using the existing techniques for similar tasks. As a demonstration case, the cancer-related gene expression of CD45^neg^/EpCAM^neg^ cells isolated from the blood samples of healthy donors and cancer patients was successfully compared. The initial results indicate that the CD45^neg^/EpCAM^neg^ nucleated cell population in the blood samples of cancer patients might contain cancer-related cells, particularly EMT-transformed CTCs, as suggested by the high detection rate of vimentin gene expression. Overall, this study presents an ODEP microfluidic system capable of simply and effectively isolating a specific, rare cell species from a cell mixture.

## 1. Introduction

Cancer metastasis is the main cause of cancer-induced death [1]. Circulating tumour cells (CTCs), which have been documented since 1869, are cells that escape from the primary tumour site into the contiguous vasculature and subsequently exist in the blood circulation [2]. Growing evidence has revealed that the presence of CTCs in the blood circulation is linked to cancer metastasis or relapse [1]. Therefore, CTC studies have great potential for understanding the mechanisms behind cancer metastasis. This information could greatly stimulate scientists to develop new therapeutic strategies for cancer treatments. In terms of clinical applications, moreover, it has been reported that the analysis of CTCs, regarded as a liquid tumour biopsy, can be utilized as a diagnostic or prognostic tool [3] for monitoring cancer metastasis or the therapeutic response [4,5] and for guiding individualized treatment [6]. To achieve these goals, it is necessary to isolate CTCs from a blood sample at high purity to possibly avoid the analytical interferences caused by the surrounding blood cells (mainly leucocytes) [7].

Nevertheless, CTCs are very rare in a blood sample, with an approximate concentration of 1 CTC per 10^5^–10^7^ blood mononuclear cells [8]. This phenomenon makes CTCs difficult to isolate and purify, particularly in a high-purity manner. Leveraging the recent advances in cell isolation techniques, a wide variety of CTC isolation and purification methods have been presented and can be broadly classified as physical and biochemical methods [9]. Overall, the physical-based methods [8,10] (mainly filtration [11,12,13]) for CTC isolation are easy to perform and do not require labelling of the harvested cells, but the cell purity is less than the purity achieved by biochemical protocols. In the biochemical-based schemes, the techniques of immunomagnetic cell separation are predominantly adopted for CTC isolation and purification [14,15]. In these methods, magnetic beads coupled to antibodies specific to a CTCs’ surface antigen (mainly the epithelial cell adhesion molecule (EpCAM) and cytokeratins (CKs)) are commonly utilized to recognize and bind the CTCs [15]. The magnetically labelled CTCs are then separated from the surrounding cells via an applied magnetic field. CTC isolation based on this strategy is primarily utilized in current CTC isolation or detection systems (e.g., the CellSearch^TM^ system [16], the magnetic-activated cell sorting system [17,18,19], or Dynabeads^TM^ [20]). Overall, the cell purity of CTCs obtained by the abovementioned cell isolation scheme ranges from 20% to 50% [16,19,20].

Although the abovementioned CTC isolation schemes have been successfully demonstrated, leucocyte contamination of the harvested CTCs is commonly unavoidable, possibly causing problems in the subsequent CTC-related analysis, particularly gene expression assays. This problem is primarily because the expression levels of some leucocyte genes are unclear. Therefore, their presence could interfere with subsequent analytical works [7]. This fact highlights the importance of isolating high-purity (ideally 100%) CTCs for the subsequent high-precision analyses. In addition to the CTC purity problem, there are some important biological issues that should be further considered. As discussed earlier, the majority of CTC isolation or purification strategies rely primarily on the use of EpCAM or CKs for the identification of CTCs. Nevertheless, CTCs, particularly those with a highly metastatic nature, might undergo the so-called epithelial-to-mesenchymal transition (EMT) [21]. Afterward, the CTCs might reduce their expression of EpCAM and CKs [22] and become motile cells for migration to distant metastatic sites [23]. In this situation, these clinically important CTCs associated with cancer metastasis might be missed if the conventional CTC isolation methodologies are used [24]. This possibility might, to some extent, explain the fact that these EMT-transformed CTCs are less studied in the literature. 

Leveraging the technical advantages of microfluidic technology, moreover, several microfluidic systems have been demonstrated for the isolation and purification of CTCs with better performance than the conventional approaches [25]. For example, the CTC-iChip [14], two-stage microfluidic chip [26], nanostructure embedded microchips [27], parallel flow micro-aperture chip [28], and herringbone chip [29] mainly utilize EpCAM or other surface antigen-specific antibodies to recognize, separate, and then isolate CTCs from the surrounding cells in the microfluidic systems. Although the isolation and purification of CTCs with higher purity (9.2 to 70% [14,26,27,28]) was achieved by utilizing these microscale devices, the problem of leucocyte contamination in the treated samples has not yet been fully solved. Otherwise, these reported devices or protocols are mainly designed for the positive isolation of CTCs (i.e., EpCAM^pos (positive)^ CTCs), ignoring the clinically important CTCs that have undergone EMT (i.e., EpCAM^neg (negative)^ CTCs).

To address the issues mentioned above, we attempted to integrate the techniques of fluorescence microscopic observations and optically induced dielectrophoretic (ODEP) force-based cell manipulation in a microfluidic system to isolate the specific cells of interest in a high-purity manner. In the design, immunofluorescence dye staining and fluorescence microscopic observations were first used to recognize the target cells. This recognition was followed by ODEP-based cell manipulation to precisely isolate the targeted cells from the surrounding cells. In this study, an ODEP microfluidic system was designed and fabricated. Moreover, the appropriate operating conditions (e.g., the size of the circular light image for manipulating a target cell, and the ring belt width of the circular ring dark image for repelling the non-targeted surrounding cells) for precisely manipulating cells were experimentally determined. In this work, the presented ODEP microfluidic system was utilized to isolate live CD45^neg^ and EpCAM^neg^ cells from blood samples that might contain CTCs that had undergone EMT. The performance of CD45^neg^/EpCAM^neg^ cell isolation based on the presented method was then experimentally evaluated using a cancer cell line model. To initially explore the clinical significance of these cells (i.e., CD45^neg^/EpCAM^neg^ cells) that are generally ignored in conventional CTC studies, a small-scale clinical experiment comparing the cancer-related gene expression levels of the CD45^neg^/EpCAM^neg^ cells isolated from the blood samples of healthy donors and head-and-neck cancer patients was carried out. 

The results revealed that the optimal diameter ratio of the target cell manipulated to the circular light image used for such cell manipulation was approximately 50% within the explored experimental conditions. In addition, the optimal ring belt width of the circular ring dark image for effective repellence of the surrounding leucocytes was 60 μm. In terms of the CD45^neg^/EpCAM^neg^ cell isolation performance, the tested cancer cell line model demonstrated that the proposed method was able to isolate CD45^neg^/EpCAM^neg^ cells without the contamination of leucocytes, the most abundant cell population in the treated cell samples. This result was further confirmed by the following clinical sample tests showing that the purity of the CD45^neg^/EpCAM^neg^ cells isolated from the blood samples of cancer patients was as high as 100%, which is currently impossible when utilizing existing techniques for similar tasks. As a demonstration case, a comparison of the cancer-related gene expression levels of the CD45^neg^/EpCAM^neg^ cells isolated from the blood samples of healthy donors and cancer patients was successfully performed. The initial results might indicate that the CD45^neg^/EpCAM^neg^ nucleated cell population in the blood samples of cancer patients might contain cancer-related cells, particularly EMT transformed CTCs, as suggested by the high detection rate of vimentin gene expression. Overall, this study has presented an ODEP microfluidic system capable of simply and effectively isolating a specific rare cell species from a cell mixture. 

## 2. Materials and Methods

### 2.1. Design of the ODEP Microfluidic System for High-Purity Isolation of CD45^neg^/EpCAM^neg^ Cells

An ODEP microfluidic system was proposed to separate, and then isolate the CD45^neg^/EpCAM^neg^ cells from the surrounding leucocytes in a batch-wise manner utilizing the techniques of fluorescence microscopic observation and ODEP force-based cell manipulation in a microfluidic system. The layout of the ODEP microfluidic system is illustrated in Figure 1a. In the study, a T-shaped microchannel was designed, in which the main microchannel (L: 25 mm, W: 1000 μm, H: 60 μm) was used for transportation of the cell suspension sample, and the side microchannel (L: 15 mm, W: 400 μm, H: 60 μm) was designed for the collection of the isolated cells. In the design, the fluorescence microscopic observation and the ODEP-based cell manipulations for the identification, and isolation of CD45^neg^/EpCAM^neg^ cells, respectively, were performed at the junction area (L: 1400 μm, W: 1000 μm, H: 60 μm) of the T-shaped microchannel, defined as the cell isolation zone. In the microfluidic system (Figure 1a), three through-holes (D: 1 mm) for tubing connections were designed, one each for loading and collecting the fresh and waste cell suspension samples, respectively, and the other one for harvesting the isolated cells. The assembly of the ODEP microfluidic system is schematically shown in Figure 1b. Briefly, the ODEP microfluidic system was composed of a top fabricated polydimethylsiloxane (PDMS) substrate (Layer A), an indium-tin-oxide (ITO) glass substrate (Layer B), double-sided adhesive tape with microfabricated microchannels (Layer C, thickness: 60 μm), and a bottom ITO glass substrate coated with a layer of photoconductive material (encompassing a 20-nm-thick n-type hydrogenated amorphous silicon layer and a 1-μm-thick hydrogenated amorphous silicon layer) (Layer D). In terms of structure, the three through-holes were located in Layer A and Layer B and were connected directly to the microchannels in Layer C. 

### 2.2. Microfabrication and Experimental Setup

The overall fabrication process was based on a computer-numerical-controlled (CNC) machining, a metal mould-punching fabrication process, a PDMS replica moulding, a thin-film technology using sputtering and plasma-enhanced chemical vapour deposition (PECVD), and a plasma oxidation-aided bonding process [30]. Briefly, the three components of the top PDMS layer (Layer A) (Figure 1b) were fabricated by the combination of CNC machining and PDMS replica moulding, as described previously [30]. For the preparation of the ITO glass substrate (Layer B) (Figure 1b), the three through-holes were mechanically drilled in an ITO glass (15 Ω, 0.7 mm; Ritek, Taiwan) using a driller (rotational speed: 26,000 rpm). For Layer C, a custom-made metal mould was used to create the hollow structure of the T-shaped microchannels in double-sided adhesive tape (9009, 3 M, Taiwan) through a manually punching operation. For the bottom substrate (Layer D) (Figure 1b), a 70-nm-thick ITO layer was first sputtered onto a cleaned dummy glass and then thermally annealed (240 °C, 60 min). A 20-nm-thick, n-type hydrogenated amorphous silicon layer was then sputtered onto the ITO layer to improve the adhesion between the fabricated ITO glass and the amorphous silicon layer to be deposited in the subsequent process. Next, a 1-μm-thick amorphous silicon layer was deposited onto the treated ITO glass through a PECVD process [30]. 

In the subsequent assembly process, the three through-hole component (Layer A) was bonded with Layer B via O_2_ plasma surface treatment. This step was followed by assembling with Layer D with the fabricated double-sided adhesive tape (Layer C) (Figure 1b). In operations, the loaded cell suspension sample was transported in the main microchannel using a suction-type syringe pump (Syringe pump 1; Figure 1c). To achieve the ODEP force-based cell manipulation [30], a function generator was used to apply an alternating current (AC) voltage between the two ITO glass layers (Layers B and D; Figure 1b) of the proposed system. A commercial digital projector (PLC-XU350, SANYO, Osaka, Japan) coupled to a computer (Computer 1; Figure 1c) was used to display light images onto the photoconductive material (Layer D) to generate an ODEP force on the manipulated cells. In addition, a CCD-equipped fluorescence microscope (Zoom 160, Optem, Fairport, NY, USA) was utilized to observe the manipulation of cells in the proposed system. The overall experimental setup is schematically illustrated in Figure 1c (a photograph of the experimental setup is provided as a Appendix A).

### 2.3. The Working Scheme for the Isolation and Purification of CD45^neg^/EpCAM^neg^ Cells

In this work, the combination of fluorescence microscopic observation and ODEP-based cell manipulation was used to identify, target, separate, and finally isolate the desired CD45^neg^/EpCAM^neg^ cells from the surrounding leucocytes in the presented ODEP microfluidic system. The overall working procedures are illustrated in Figure 2. Briefly, all cells in a treated cell suspension were first stained with different fluorescent dyes to identify EpCAM surface marker-positive CTCs (green colour dot images), CD45 surface marker-positive leucocytes (green colour dot images), calcein AM-positive live cells (orange colour dot images), and Hoechst-positive nucleated cells (blue colour dot images). The treated cell suspension sample was then loaded in the ODEP microfluidic system, and flowed through the main microchannel (Figure 2a). During this operation, fluorescence microscopic observation was simultaneously carried out at the defined cell isolation zone to detect the desired live CD45^neg^/EpCAM^neg^ cells (i.e., the orange colour-only dot images). Briefly, the optical filter in the fluorescence microscope was set to observe the green and orange colour dot images. Once an orange colour-only dot image was observed, the cell suspension flow was stopped (Figure 2b). In the following procedures for the identification and positioning of the desired cells, the optical filters were switched to first observe the green colour-only dot images (i.e., the leucocytes or EpCAM^pos^ CTCs) (Figure 2c) and then the blue colour-only dot images (i.e., all nucleated cells) (Figure 2d). After the contradistinction, the desired cell (i.e., the live EpCAM^neg^, CD45^neg^ and Hoechst^pos^ cells) in the cell suspension was then targeted under light field microscopy imaging (Figure 2e). This step was followed by performing ODEP-based cell manipulation to isolate the cell targeted from the surrounding cells (Figure 2f–i). Briefly, a large-area light was used to illuminate the cell isolation zone to attract and then anchor the cells within the light-projected region, as illustrated in Figure 2f. Meanwhile, a circular ring dark image (OD: 40 μm; ID: 20 μm) was first utilized to enclose the targeted cell and soon followed by expansion of the width of the ring belt (OD: 80 μm; ID: 20 μm), as shown in Figure 2f,g. In this work, the circular ring dark image, functioning as a protective ring belt, was designed to prevent the unwanted adhesion of the surrounding cells to the manipulated cells during the operation process. The circular ring dark image around and the circular light image on the targeted cell were then moved to deliver the enclosed cell to the side microchannel, as illustrated in Figure 2g–i. After repeating the above processes, the cells of interest (i.e., the live CD45^neg^/EpCAM^neg^ cells) could be isolated, purified, and finally collected in the side microchannel for the subsequent collection using a suction-type syringe pump (i.e., Syringe pump 2; Figure 1c).

### 2.4. Working Mechanism and the Optimal Operating Conditions for the ODEP Force-Based Cell Manipulation in This Work

The ODEP-based microparticle manipulation first presented in 2005 [31] has been well described elsewhere [7,30,32]. Briefly, an AC electrical voltage is first applied between the two ITO glasses (e.g., Layer B and D; Figure 1b) to produce a uniform electric field between the two glass substrates. Under this circumstance, the electrically neutral microparticles (e.g., biological cells) within such an electric field become electrically polarized. The projection of light onto the photoconductive material on the bottom ITO glass (e.g., Layer D; Figure 1b) can lead to a decrease in the electrical voltage across the liquid layer within the light-illuminated region. This can thus generate a non-uniform electric field within the ODEP system. In the ODEP force-based microparticle manipulation, the interaction between the non-uniform electric field and electrically polarized particles is used to manipulate the microparticles [30]. In practical applications, one can simply control the movements of a light image to manipulate microparticles in a manageable manner. 

In the proposed ODEP microfluidic system, the ODEP-based cell manipulation was utilized to separate and then isolate CD45^neg^/EpCAM^neg^ cells from the surrounding cells, as schematically illustrated in Figure 2. In this work, the optimal operating conditions (e.g., the size of the circular light image for attracting and dragging the enclosed target cell and the ring belt width of the circular ring dark image for repelling the non-targeted surrounding cells, as illustrated in Figure 2f–i) were experimentally evaluated. In this study, the ODEP force acting on a cell was evaluated based on a method described previously [30]. Briefly, the ODEP manipulation force, which is the net force between the ODEP force and the friction force, generated on the manipulated cell was experimentally evaluated in this work. In a steady state, the ODEP manipulation force of a cell is balanced by the viscous drag of the fluid. Therefore, the hydrodynamic drag force of a moving cell is normally used to evaluate the net ODEP manipulation force of a cell according to Stokes’ law (Equation (1)) [33] for describing the drag force (*F*) exerted on a spherical particle in a continuous flow condition.
*F* = 6π*rηv*(1)
where *r*, *η*, and *v* denote the radius of the cell, the viscosity of the fluid, and the terminal velocity of the cell, respectively [30]. According to Stokes’ law, therefore, the ODEP manipulation force can then be experimentally evaluated through the measurement of the maximum velocity of a moving light image that can manipulate a cell, as discussed previously [30].

Moreover, the ODEP force generated on a cell can be theoretically expressed by Equation (2), which was also used to describe the dielectrophoresis (DEP) force [34]:
*F_DEP_* = 2*πr*^3^*ε_m_*Re[*K*(*ω*)]∇*E*^2^(2)
where *r*, *ε_m_*, *K*(*ω*), and ∇*E*^2^ denote the cell radius, the permittivity of the solution surrounding the cells, the Clausius–Mossotti factor (associated with the frequency of the electric field, the conductivity of the medium, the internal conductivity of the cells, and the membrane capacitance of the cells [35]), and the gradient of the electric field squared, respectively [36]. In terms of ODEP operating conditions, overall, the ODEP force generated on a specific cell is influenced by the magnitude and frequency of the applied electrical voltage under a given solution condition. Therefore, the electrical properties used in this work were first measured to determine the conditions under which no cell aggregation phenomenon occurs. In addition to the electrical conditions for ODEP operations, the size of the light image is also reported to influence the ODEP-based microparticle manipulation [37]. In this work, experimental evaluations were performed to determine the quantitative relationship between the diameter of the circular light image (20, 30, 40 μm) and the maximum velocity (μm s^−1^) of the moving light image that can manipulate (i.e., attract and drag) PC-3 and SW620 cancer cells. Similar work was also carried out to evaluate the effect of the width of the dark image bar (20, 40, 60, 80 μm) on the manipulation (i.e., repulse and push) of leucocytes. After these evaluations, the optimal conditions for the light or dark image were determined for the ODEP-based cell manipulation as described in Figure 2. 

### 2.5. Evaluation of the CD45^neg^/EpCAM^neg^ Cell Isolation Performance-Cancer Cell Line Model

Experiments were conducted to evaluate the performance of the presented ODEP microfluidic system for CD45^neg^/EpCAM^neg^ cell isolation. This study was approved by the Institutional Review Board of the Chang Gung Memorial Hospital. Informed consent was obtained from all blood sample donors (Approval ID: 201601081B0). All the methods were conducted in accordance with the relevant guidelines for clinical tests. Briefly, blood samples (4 mL each) were obtained from 3 healthy donors. The erythrocytes in the blood samples were first lysed using an erythrocyte lysis buffer [38]. The treated sample was further processed to deplete (CD45^pos^) leucocytes using the EasySep Human CD45 Depletion Kit (StemCell Technologies, Vancouver, BC, Canada). After erythrocyte and leucocyte depletion, the cells remaining in the treated samples were stained with fluorescent dyes for identifying EpCAM surface marker-positive CTCs (green colour dot images) (Alexa Fluor 488-labelled EpCAM antibody, Cell Signalling Technology, Danvers, MA, USA), CD45 surface marker-positive leucocytes (green colour dot images) (FITC-labelled CD45 antibody, eBioscience, San Diego, CA, USA), calcein AM-positive live cells (orange colour dot images) (Thermo Fisher Scientific, Waltham, MA, USA), and Hoechst-positive nucleated cells (blue colour dot images) (Thermo Fisher Scientific). In this work, SW620 (cancer cell line) cells were spiked into the abovementioned treated cell samples to mimic the existence of CD45^neg^/EpCAM^neg^ cells in the cell samples. According to the previous report [39], there are about 10^2^ to 10^4^ CD45neg/EpCAMneg cells/mL in the blood samples of head-and-neck cancer patients. Based on this result, therefore, we spiked 2000 cells to 4 mL of blood sample from healthy donor (i.e., 500 CD45^neg^/EpCAM^neg^ cells/mL). In general, there will be about 2 × 10^7^ cells remained in the sample after 4 mL of blood sample is treated with blood cell depletion processes. Therefore, the percentage of the CD45^neg^/EpCAM^neg^ cells spiked in a treated blood sample was about 0.01% (2000/2 × 10^7^ × 100%). Briefly, 2000 SW620 cancer cells stained with the abovementioned fluorescent dyes except for the EpCAM surface marker dye were spiked into the treated cell samples. The treated cell samples were then prepared in 100 μL of 300 mM sucrose solution (solution conductivity: 3.4 μS cm^−1^). This step was soon followed by the ODEP-based CD45^neg^/EpCAM^neg^ cell isolation described in Figure 2. The gene expression of the harvested CD45^neg^/EpCAM^neg^ cells was then analysed using the real-time polymerase chain reaction (PCR) technique. Briefly, the total RNA of the cells was extracted using the PicoPure RNA isolation Kit (Thermo Fisher Scientific) and the cDNA was then synthesized via the SuperScript IV First-Strand Synthesis System (Thermo Fisher Scientific). This step was followed by cDNA pre-amplification using TaqMan PreAmp Master Mix (Thermo Fisher Scientific) to increase the quantity of the target genes. Afterward, TaqMan-based detection was carried to determine the gene expression levels of the isolated cells. In this work, the TaqMan assays for each gene were purchased from Thermo Fisher Scientific and were performed according to the manufacturer’s instructions. The β-2-microglobulin serves as the internal control gene in this study.

### 2.6. Comparison of the Cancer-Related Gene Expression of the CD45^neg^/EpCAM^neg^ Cell Populations Isolated from the Blood Samples of Healthy Donors and Head-and-Neck Cancer Patients

To initially explore the clinical significance of the CD45^neg^/EpCAM^neg^ cell populations in the blood samples of cancer patients, the following small-scale clinical test was carried out. Briefly, blood samples (4 mL each) were obtained from patients diagnosed with head-and-neck cancer (*n* = 8) and from healthy blood donors (*n* = 5). The blood samples were then processed using the protocol described earlier to isolate the CD45^neg^/EpCAM^neg^ cell population. In this study, we only harvested about 50 CD45^neg^/EpCAM^neg^ cells in a blood sample for the subsequent gene expression analysis. This is mainly because 25–50 cells were technically sufficient for the subsequent analysis of their gene expression. The harvested cells were then analysed to determine their cancer-related gene expression using real-time PCR as described earlier.

### 2.7. Statistical Analysis

Data from at least three separate experiments were analysed and presented as the mean ± the standard deviation. One-way analysis of variance (ANOVA) was used to examine the effect of the experimental conditions on the results. The Tukey honestly significant difference (HSD) post hoc test was used to compare the differences between two investigated conditions when the null hypothesis of ANOVA was rejected.

## 3. Results and Discussions

### 3.1. Characteristic Features of the Proposed ODEP-Based Microfluidic System for the Isolation and Purification of CD45^neg^/EpCAM^neg^ Cells

In this study, the integration of fluorescence microscopic observation and ODEP force-based cell manipulation in a microfluidic system was proposed for high-purity isolation of CD45^neg^/EpCAM^neg^ cells based on the working schemes described in Figure 2. First, the presented ODEP microfluidic system features in providing a gentler process for separating and isolating viable cells from a cell mixture than current techniques [30,40]. This technical advantage might be difficult to achieve using other microfluidic systems designed for the same purpose, in which the isolated cells might be damaged due to the high fluidic shear stress in a microfluidic system. This characteristic is found to be valuable for using the harvested viable cells for subsequent gene expression analysis. Additionally, in terms of the cell manipulation technique, a more user-friendly and flexible ODEP force-based working mechanism was adopted in this design compared to that in the other methods (e.g., techniques based on fluidic control [40], magnetic force [14], thermal control [41], or dielectrophoretic force (DEP) [42]) reported in the literature. Among the cell manipulation methods, the DEP force-based mechanism has been actively proposed for a wide variety of applications [43], mainly due to its capability of providing precise cell manipulation and control. Nevertheless, the DEP-based method requires a technically demanding, costly, and lengthy microfabrication process to create a unique metal microelectrode array on a substrate that is specific to the application. This technical shortcoming can be adequately solved by using the ODEP force-based technique, by which a specific microelectrode layout can be simply and flexibly created or modified through the manipulation of the optical patterns projected on the photoconductive material-coated ITO glass of the ODEP system (e.g., Figure 1c) that act as a virtual electrode layout [31]. In practice, one can simply utilize a commercial digital projector to display optical patterns on the ODEP system to manipulate cells in a simple, flexible, and user-friendly manner via computer-interface controls [30]. 

In terms of cell-related studies, the utilization of the ODEP-based mechanism in microfluidic systems has been presented for various applications, mainly cell patterning [31], cell lysis [44], cell fusion [45], cell sorting based on the cellular size [30] or electrical properties [46], and cell isolation and purification [7]. For the last example, ODEP-based methods have been presented for the isolation of bacteria [47], droplets [48], and CTCs [7]. For CTC isolation, the ODEP technique was found to be particularly promising for the isolation of rare cells in a cell mixture in a high-purity manner due to its ability for precise cell manipulation [7]. The level of cell isolation achieved by such a methodology is normally technically impossible by the conventional macro-scale devices or methods [14,16,20]. In operations, light images are commonly used to target the wanted cells, and such light images are then used to manipulate and then separate the targeted cells from the surrounding cells for cell isolation and purification purposes [7]. It was recently reported that the CTC isolation techniques based on this method can achieve a CTC cell purity as high as 100% [7], which is currently impossible using the other existing methods. During the cell manipulation process using moving light images, however, cautious operations are normally required to prevent the light images from attracting unwanted cells around the target cells [7]. Otherwise, the purity of the harvested cells might be seriously affected. To address this issue, one of the technical features of the presented method is to use a large-area light to illuminate the cell isolation zone to attract and then immobilize all cells within this zone on the surface of a substrate (Figure 2f). Afterward, a circular ring dark image, functioning as a protective ring belt, was designed to enclose the targeted cell, effectively preventing the unwanted adhesion of the surrounding cells to the manipulated cells during the target cell isolation process, as described in Figure 2f–h. This design not only provides high-purity isolation of the target cells but also makes the cell manipulation process simpler and more user-friendly.

### 3.2. Optimal Operating Conditions for ODEP Force-Based Cell Manipulation

In this study, ODEP-based cell manipulation was utilized to separate and then isolate the CD45^neg^/EpCAM^neg^ cells from the surrounding cells, as illustrated in Figure 2. For effective and efficient cell isolation, the optimal operating conditions were experimentally evaluated. As described earlier in Equation (2), the ODEP force generated on a specific cell is influenced by the magnitude and frequency of the electrical voltage applied under a given solution condition. In this study, the electrical condition of 8 V (3 MHz) was first evaluated experimentally, and no cell aggregation phenomenon was observed under such condition. This result could greatly facilitate the subsequent cell isolation process. To determine the optimum size of light and dark images for cell manipulation, experimental evaluations were also performed. In this work, two kinds of cancer cell lines with different sizes (microscopically estimated diameter: 20.1 ± 1.5 and 10.5 ± 0.9 μm for PC-3 and SW620 cancer cells, respectively) were used as model cells for the testing. In this study, the ODEP manipulation force acting on a cell was evaluated through measurement of the maximum velocity of a moving light (dark) image that can manipulate a cell, as discussed previously [7]. In this evaluation, the quantitative relationship between the diameter (20, 30, 40 μm) of the circular light image that was used (i.e., the light illuminating the targeted cell; Figure 2f–i) and the maximum velocity (μm s^−1^) of the moving light image that can manipulate (i.e., attract and drag) PC-3 and SW620 cancer cells was first established. Similar work was also carried out to evaluate the effect of the dark bar image width (20, 40, 60, 80 μm) on the manipulation (i.e., repulse and push) of the surrounding cells (mainly leucocytes). 

The results (Figure 3a,b) revealed that the diameter of the circular light image had a significant influence (*p* < 0.01, ANOVA) on the maximum velocity (μm s^−1^) of the moving light image that can manipulate the SW620 and PC-3 cancer cells. This result was in line with the previous findings demonstrating that the size of a light image can play a role in the ODEP-based microparticle manipulation [37]. For the SW620 cancer cells with average diameter of 10.5 ± 0.9 μm, the measured maximum velocity that could manipulate this cell type significantly (*p* < 0.01) decreased when the largest circular light image (D: 40 μm) was used with the explored experimental conditions (Figure 3a). This finding can be explained by the fact that under an ODEP field, the electric field within a smaller light image could be more focused than that in a larger image. This feature could accordingly contribute to a higher ODEP manipulation force and thus higher maximum velocity for the light image manipulation of a cell [7]. For the PC-3 cancer cells with an average diameter of 20.1 ± 1.5 μm, conversely, the measured maximum velocity significantly (*p* < 0.05) decreased when the smallest circular light image (D: 20 μm) was used (Figure 3b). This unexpected result could be because the circular light image with the smallest size (D: 20 μm) was close to the average size (20.1 ± 1.5 μm) of the PC-3 cancer cells used in this study. This similarity could, therefore, lead to instability in the ODEP force generation, as discovered previously [7]. 

In terms of the ratio of the cell diameter to the circular light image diameter, overall, the maximum velocity of the circular light image that could manipulate a cell was higher when the ratio was in the range of 35.1–52.7% and 50.2–66.9% for the SW620 and PC-3 cancer cells, respectively (Figure 3a,b). Based on these results, the ratio was set at 50% for the following experiments. In this study, moreover, a circular ring dark image was designed to enclose the abovementioned circular light image when a cell was targeted. As described earlier, this design was to repel the surrounding cells (mainly leucocytes) and thus, to prevent the unwanted adhesion of leucocytes to the manipulated cells during the target cell isolation process (Figure 2f–i). In this study, a similar evaluation was carried out to determine the optimal ring belt width of the circular ring dark image for the manipulation (i.e., repulse and push) of the leucocyte. The results (Figure 3c) exhibited that the size of the ring belt width had a significant (*p* < 0.01; ANOVA) impact on the leucocyte manipulation, in which the maximum velocity of the dark bar image that could manipulate a leucocyte increased as the dark bar image width increased from 20 μm to 60 μm. According to the results (Figure 3c), therefore, the ring belt width of the circular ring dark image was set at 60 μm for the following works. 

### 3.3. Performance Evaluation of CD45^neg^/EpCAM^neg^ Cell Isolation

In this study, the performance of the proposed ODEP microfluidic system for the isolation of CD45^neg^/EpCAM^neg^ cells was experimentally assessed. First, the erythrocytes and leucocytes in the blood samples of healthy donors were mostly depleted. The remaining cells in the treated samples were then stained with four kinds of fluorescent dyes, as described in Figure 2. In this work, SW620 cancer cells stained with the four fluorescence dyes except for the EpCAM antibody-based dye were spiked into the abovementioned treated cell samples. Although SW620 cancer cells normally have surface antigen expression of EpCAM in nature, these cells were not stained with the EpCAM antibody-based dye in this study, making them non-observable in the fluorescence microscopic observation for the identification of CD45^pos^ or EpCAM^pos^ cells, as described in Figure 2a–c. Based on this design, therefore, the SW620 cancer cells spiked in a sample would be grouped into the CD45^neg^/EpCAM^neg^ cell population in this study. This design was to mimic the existence of cancer-related cells in the CD45^neg^/EpCAM^neg^ cell population in the cell samples. Afterward, the live CD45^neg^/EpCAM^neg^ nucleated cells were isolated from the cell samples based on the procedures described in Figure 2. The recovery rate of the device was evaluated to be 81.0 ± 0.7%. In this study, 50 cells were isolated and delivered to the side microchannel for the subsequent collection. Practically, about 40 cells were obtained after the subsequent collection process. The harvested live CD45^neg^/EpCAM^neg^ nucleated cells were then analysed to determine their expression of leucocyte-specific (e.g., CD45) and cancer cell-specific (e.g., EpCAM, and CK19) genes using a real-time PCR system. 

The results (Figure 4a) revealed that the expression of the CD45 gene was detectable only in pure leucocytes (as a control) but not in pure SW620 cancer cells, and the CD45^neg^/EpCAM^neg^ nucleated cells harvested via the proposed cell isolation scheme. This result demonstrated that the CD45^neg^/EpCAM^neg^ nucleated cells harvested through the proposed ODEP microfluidic system were pure and not contaminated with leucocytes, the most abundant cell population in the treated cell samples. This technical advantage was found to be valuable for using the harvested cells for the subsequent high-precision gene expression analysis because the existence of the surrounding cells, such as leucocytes, could cause analytical interference [7]. Moreover, the expression of the EpCAM and CK19 genes was detected only in the harvested live CD45^neg^/EpCAM^neg^ nucleated cells but not in leucocytes (as a control) (Figure 4b,c). This result indicated that the spiked SW620 cancer cells mimicking the existence of cancer-related cells in the CD45^neg^/EpCAM^neg^ cell population were able to be successfully isolated and detected by the proposed cell isolation process, and the protocol for gene expression analysis, respectively.

In addition to the cancer cell line model described above, the performance of CD45^neg^/EpCAM^neg^ cell isolation was also evaluated using a clinical sample model. In this work, blood samples were obtained from patients diagnosed with head-and-neck cancer. The samples were first treated, and the CD45^neg^/EpCAM^neg^ cell isolation process was then performed as described in Figure 2. Figure 5 shows the real cell isolation operation process. Briefly, Figure 5a(I–III) demonstrated the process of fluorescence microscopic observation to identify the CD45^neg^/EpCAM^neg^ cells marked by the arrow; these cells were found in both the orange colour-only dot image (Figure 5a(I–II)) and the blue colour dot image (Figure 5a(III)). After the CD45^neg^/EpCAM^neg^ cell was positioned using light field microscopy imaging (Figure 5a(IV)), ODEP-based cell manipulation was carried out to isolate the cells targeted to the side microchannel (Figure 5a(V–VII)). Through repetition of the processes abovementioned, the CD45^neg^/EpCAM^neg^ cells were able to be isolated and temperately stored in the side microchannel for the subsequent collection. In this work, the purity of the live CD45^neg^/EpCAM^neg^ cells in the side microchannel was further evaluated microscopically. The results (Figure 5b) revealed that the purity of the live CD45^neg^/EpCAM^neg^ cells was as high as 100%, which is beyond the currently possible purity obtained by using other existing cell isolation techniques. In this proof-of-concept study, the proposed ODEP-based cell isolation system has not yet been automated. The operating time for processing 4 mL of blood sample was about 4 h, in which the manual fluorescent microscopic observation consumed most of the time required. As a result, the automation of the proposed system will be our future goal, which can largely improve the working throughput of the proposed ODEP-based cell isolation system.

### 3.4. Comparison of the Cancer-Related Gene Expression of the CD45^neg^/EpCAM^neg^ Cells Isolated from the Blood Samples of Healthy Donors and Head-and-Neck Cancer Patients

In conventional CTC-related studies, the cellular proteins EpCAM and CKs are predominately used as biomarkers to identify CTCs [15]. However, growing evidence has suggested that the use of these biomarkers to identify CTCs is not sufficient due to the heterogeneous characteristics of CTCs [24]. As a result, the search for other more clinically meaningful markers that might be more relevant to cancer metastasis has become important. It is well recognized that the CTCs with a highly metastatic nature might undergo EMT [21], after which their expression of EpCAM and CKs is downregulated [22]. Therefore, these cells are generally ignored in the conventional positive selection-based CTC isolation schemes. In our recent study, moreover, it was discovered that the number of CD45^neg^/EpCAM^neg^ nucleated cells in the blood samples of cancer patients is significantly higher than that of healthy donors [49]. To initially explore the clinical significance of the CD45^neg^/EpCAM^neg^ nucleated cells, in which the EMT transformed CTCs might contain, a preliminary clinical test was carried out. In this work, the CD45^neg^/EpCAM^neg^ nucleated cells were isolated from the blood samples of patients diagnosed with head-and-neck cancer and from healthy donors based on the processes described in Figure 2. The cancer-related gene expression of the CD45^neg^/EpCAM^neg^ nucleated cells in the two sample groups was analysed and then compared. In this work, the investigated cancer-related genes were genes related to EMT (CDH1, CDH2, EpCAM, CK19, Vimentin, SNAIL1, TWIST1) [50], multiple drug resistance (MRP1, MRP2, MRP4, MRP5, MRP7) [51], and cancer stem cell (CSC) (ALDH1, NANOG, OCT4, SOX2, CD133) [52]. 

Table 1 provides a summary of the results of this comparison (the detailed experimental results are provided as a Appendix A). In terms of the EMT-related gene expression (Table 1), briefly, EpCAM gene expression was not detected in the two compared sample groups because EpCAM^pos^ cells were excluded in the proposed CD45^neg^/EpCAM^neg^ cell isolation scheme. This result further confirmed that the purity of the CD45^neg^/EpCAM^neg^ cells isolated was high, consistent with the findings in Figure 5. Expression of the CK19 (i.e., gene for an epithelial marker) and SNAIL1 (i.e., gene for a zinc finger transcription factor that drives EMT progression [53]) genes was not detected in the healthy blood sample group but was detected in 1 of 8 cancer patients (i.e., detection date: 12.5%). Vimentin is the major protein constituent of the intermediate filament responsible for the maintenance of the cytoskeleton structure of a cell. It has been reported that vimentin is involved in tumour progression, cell migration, and tumour invasion [54]. Over-expression of vimentin in diverse tumours has been discovered previously [55,56]. In addition, vimentin expression in CTCs was also reported [57,58], and the detection of vimentin-positive CTCs has been associated with disease progression [59,60]. However, the detection of vimentin gene expression in the CD45^neg^/EpCAM^neg^ cells has not yet been reported. In this work, vimentin gene expression was detected in 1 of 5 blood samples from the healthy donors (20%) and in 6 of 8 blood samples from the cancer patients (75%). Apart from the gene expression discussed above, the expression of other evaluated EMT-related genes, such as TWIST1, CDH1, and CDH2, was undetectable in the two sample groups. Taken together, the results described above might indicate that EMT-transformed CTCs might exist in the CD45^neg^/EpCAM^neg^ cell population.

The gene expression of multi-drug resistance proteins (MRPs) including MRP1, MRP2, MRP4, MRP5, and MRP7 was not detected in the CD45^neg^/EpCAM^neg^ nucleated cells isolated from the blood samples of healthy donors (Table 1). In addition, the expression of these genes, except for the MRP1 and MRP5 genes, also was not detected in the cells isolated from the blood samples of cancer patients. Expression of the MRP1 and MRP5 genes was detected in 3 and 2 of 8 samples from cancer patients (i.e., detection rate: 37.5% and 25.0%, respectively), respectively. The MRPs belong to the family of adenosine triphosphate -binding cassette transporters and are thought to be responsible for transporting nutrients, metabolites or drugs through a cell [51]. In cancer, the over-expression of MRPs is associated with cancer cell resistance to anticancer drugs [51]. Therefore, monitoring the MRP gene expression of cancer cells could provide valuable information relevant to the disease status during or after therapies [51]. In this work, the MRP gene expression in the CD45^neg^/EpCAM^neg^ nucleated cells was studied for the first time. However, the correlation between the MRP gene expression of the CD45^neg^/EpCAM^neg^ nucleated cells and the treatment response or disease progression of cancer patients was not explored. 

Cancer stem cells (CSCs) have been found in many types of cancer, including head-and-neck cancer [61]. Cancer cells with the characteristics of stem cells are reported to be associated with cancer progression and treatment resistance [62]. Therefore, the evaluation of CSC-related gene expression in CD45^neg^/EpCAM^neg^ cells might provide information relevant to the cancer disease state. In this work, the expression of CSC-related genes (e.g., ALDH1A1, CD133, NANOG, OCT4, and SOX2) was analysed in the isolated CD45^neg^/EpCAM^neg^ nucleated cells. The results (Table 1) exhibited that only two (i.e., NANOG and OCT4) of these genes were detected in these cells isolated from the blood samples of cancer patients (2 and 4 of 8 samples, respectively), although the expression of these two genes was also found in 1 of 5 blood samples of healthy donors. It seemed that the detection rate of NANOG and OCT4 gene expression in the cancer patient group (25%, and 50%, respectively) was slightly higher than that in the healthy blood donor group (20%). In this study, as a whole, the EMT-, MRP-, and CSC-related gene expression of the CD45^neg^/EpCAM^neg^ nucleated cells isolated from the blood samples of cancer patients and healthy donors was reported for the first time. The overall results described above might indicate that the CD45^neg^/EpCAM^neg^ nucleated cells isolated from the blood samples of cancer patients might contain cancer-related cells (e.g., EMT-transformed CTCs due to the high detection rate of vimentin gene expression). Further clinical study on a large scale is required to confirm the results reported in this work. Additionally, a clinical study that links the correlations between the cancer-related gene expression of CD45^neg^/EpCAM^neg^ nucleated cells isolated from the blood samples of cancer patients and the cancer disease status might be worthwhile. 

## 4. Conclusions

The current research of CTCs or cells relevant to cancer metastasis (e.g., the CD45^neg^/EpCAM^neg^ cells in this study) is normally hindered by the lack of an adequate cell isolation method capable of efficiently and effectively harvesting these rare cell species with high purity. To address this issue, this study proposed the integration of the techniques of fluorescence microscopic observations and ODEP-based cell manipulation in a microfluidic system to isolate the specific cells of interest in a high-purity manner. In the design, the immunofluorescence dye staining and fluorescence microscopic observations were first used to recognize the targeted cells. This step was followed by ODEP-based cell manipulation to precisely isolate the targeted cells from the surrounding cells. Compared with the other microfluidic systems used for the similar tasks, the proposed ODEP microfluidic system mainly provides a way to harvest cells in a high-purity, cell-friendly, and low-cost manner. Moreover, due to the specific design of light images (i.e., circular ring dark image), the presented system provides a method to effectively prevent the unwanted adhesion of the surrounding cells to the manipulated cells during the target cell isolation process. This design not only provides higher purity isolation of the target cells but also makes the cell manipulation process simpler and more user-friendly than other ODEP-based microfluidic systems used for the same purpose. In this study, the optimal size of optical images (i.e., 50% for the diameter ratio of the manipulated target cell to the circular light image used for such cell manipulation, and 60 μm for the ring belt width of the circular ring dark image) for ODEP-based cell manipulation was experimentally evaluated. In the cancer cell line model, the proposed method was proved to be able to isolate CD45^neg^/EpCAM^neg^ cells without the contamination of leucocytes, the most abundant cell population in the treated cell samples. This performance evaluation was further confirmed by the subsequent clinical sample tests demonstrating that the purity of the CD45^neg^/EpCAM^neg^ cells isolated from the blood samples of metastatic cancer patients was as high as 100%, which is currently impossible via the existing techniques for similar tasks. In the comparison of the cancer-related gene expression in the CD45^neg^/EpCAM^neg^ cells isolated from the blood samples of healthy donors and cancer patients, the initial results indicate that the CD45^neg^/EpCAM^neg^ nucleated cell population might contain cancer-related cells, particularly EMT-transformed CTCs, as suggested by the high detection rate of vimentin gene expression. Further clinical study on a large scale is required to confirm the results reported in this work. 

## Figures and Tables

**Figure 1 micromachines-09-00563-f001:**
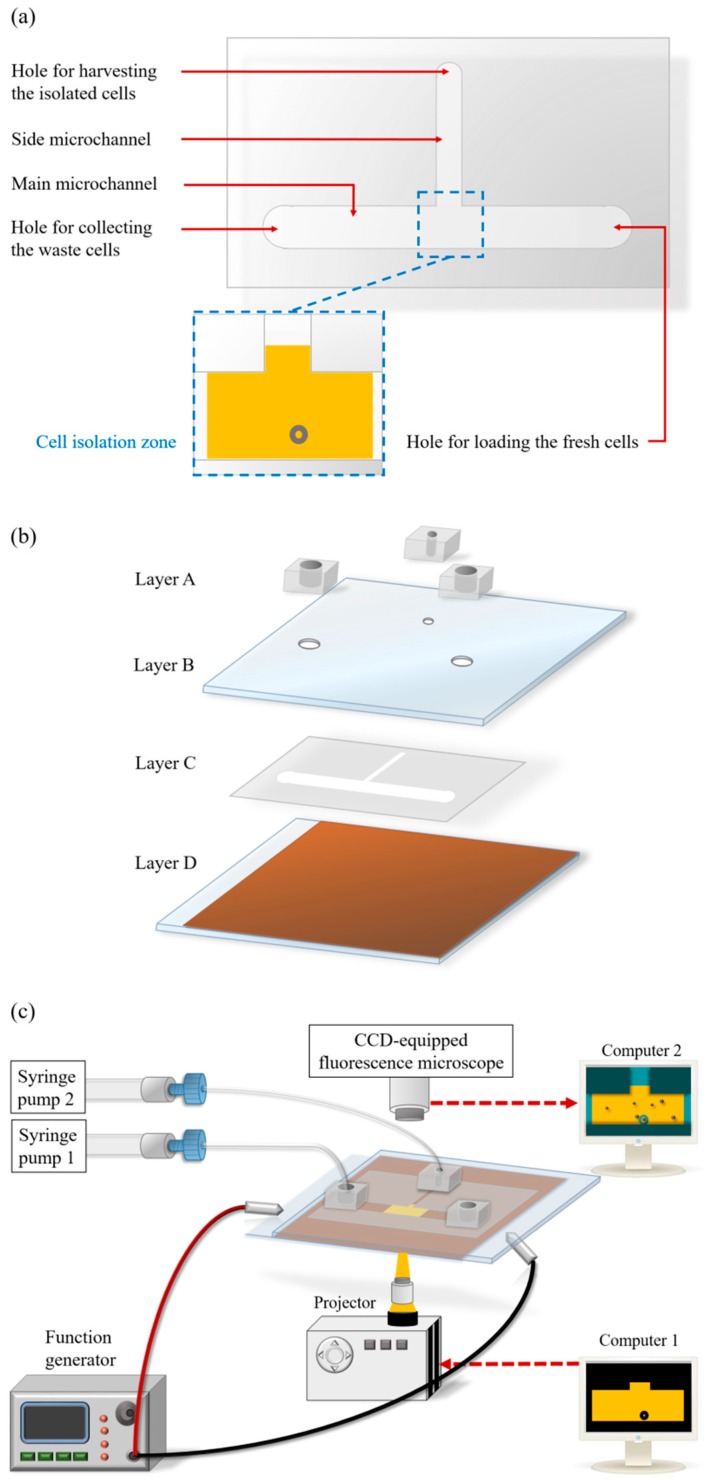
Schematic illustration of the (**a**) top-view layout and (**b**) assembly of the optically induced dielectrophoresis (ODEP) microfluidic system (Layer A: fabricated polydimethylsiloxane (PDMS) components; Layer B: indium-tin-oxide (ITO) glass; Layer C: double-sided adhesive tape with microfabricated microchannels; Layer D: ITO glass substrate coated with a layer of photoconductive material), and the (**c**) overall experimental setup.

**Figure 2 micromachines-09-00563-f002:**
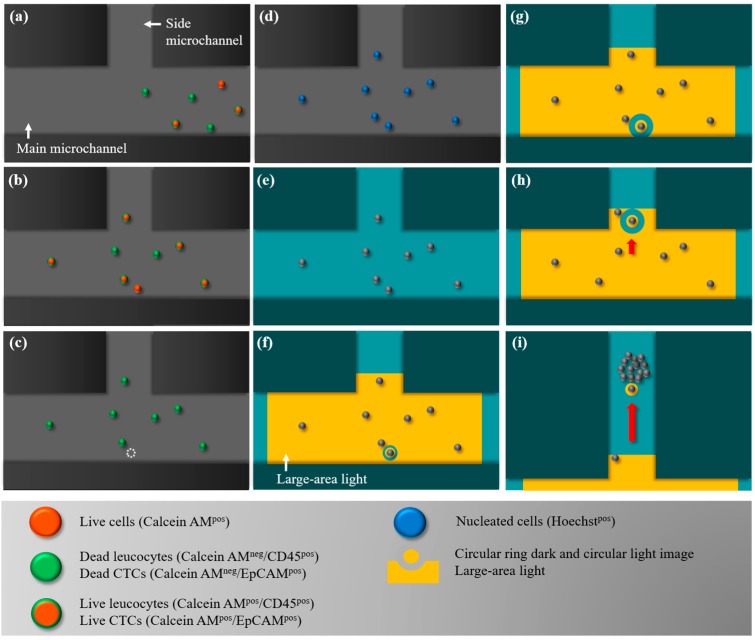
Schematic illustration of the overall working procedures for the isolation of EpCAM^neg^ and CD45^neg^ cells: (**a**) Fluorescence microscopic observation was performed at the defined cell isolation zone to detect the live EpCAM^neg^ and CD45^neg^ cells (i.e., the orange colour-only dot images) in a dynamic cell suspension flow; (**b**) the cell suspension flow was temporarily suspended when a live EpCAM^neg^ and CD45^neg^ cell (pointed out by the arrow marker) was observed in the cell isolation zone; (**c**) the optical filters in the fluorescence microscope were then switched to observe the green colour-only dot images (i.e., the leucocytes or EpCAM^pos^ CTCs); and (**d**) then the blue colour-only dot images (i.e., the all nucleated cells); (**e**) after the contradistinction, the desired cell (i.e., the live EpCAM^neg^, CD45^neg^ and Hoechst^pos^ cells) in the cell suspension was then targeted under light field microscopy (pointed out by the arrow marker); and (**f**) a large-area light was used to illuminate the cell isolation zone to anchor the cells within the light-projected region. Additionally, a circular ring dark image was utilized to first enclose the targeted cell and was (**g**) soon followed by expansion of the width of the ring belt. (**h**,**i**) The circular ring dark image around and the circular light image on the targeted cell were then moved to deliver the enclosed cell to the side microchannel for the subsequent collection.

**Figure 3 micromachines-09-00563-f003:**
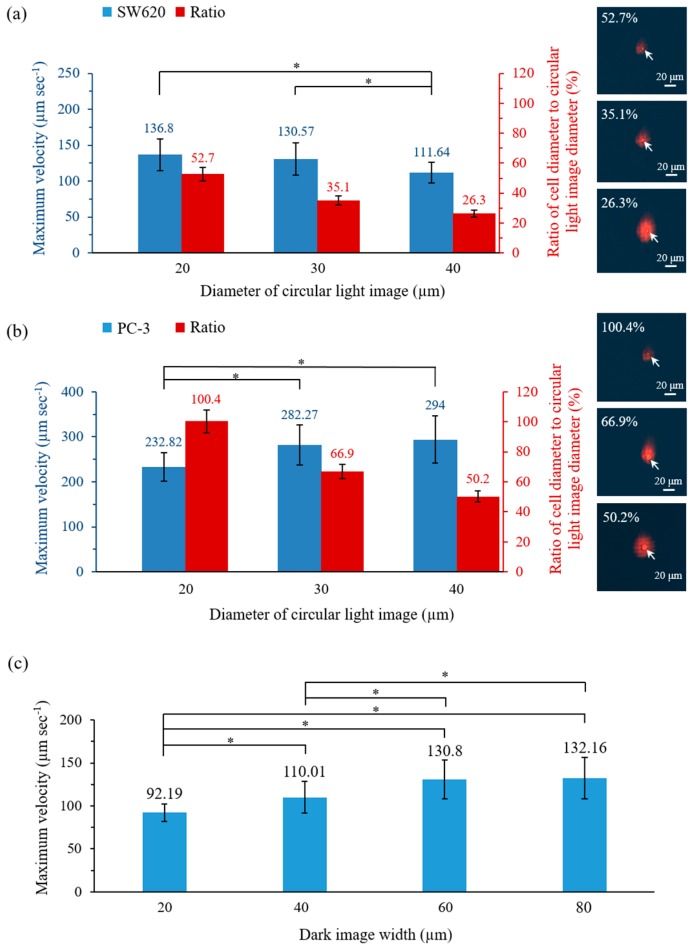
Quantitative relationship between the diameter of the circular light image (and the diameter ratio of the cell to the circular light image) that is used and the maximum velocity (μm s^−1^) of the moving circular light image that can manipulate (i.e., attract and drag) (**a**) SW620 and (**b**) PC-3 cancer cells (right column: the microscopic images of various diameter ratios of cell to circular light image; the cells are pointed out by arrow markers) and (**c**) the quantitative link between the dark image width that is used and the maximum velocity (μm s^−1^) of the moving dark image that can manipulate (i.e., repulse and push) leucocytes.

**Figure 4 micromachines-09-00563-f004:**
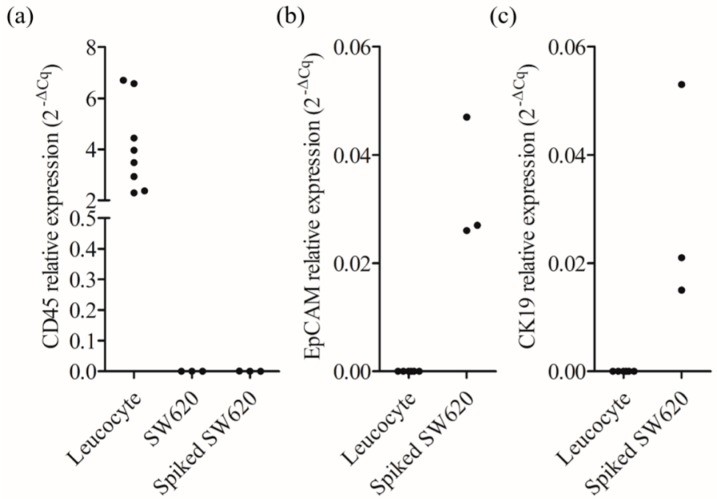
Analysis of the gene ((**a**) leucocyte-specific CD45 and cancer cell-specific (**b**) EpCAM and (**c**) CK19 genes) expression levels in the pure leucocytes, the SW620 cancer cells, and the cells isolated by the proposed method.

**Figure 5 micromachines-09-00563-f005:**
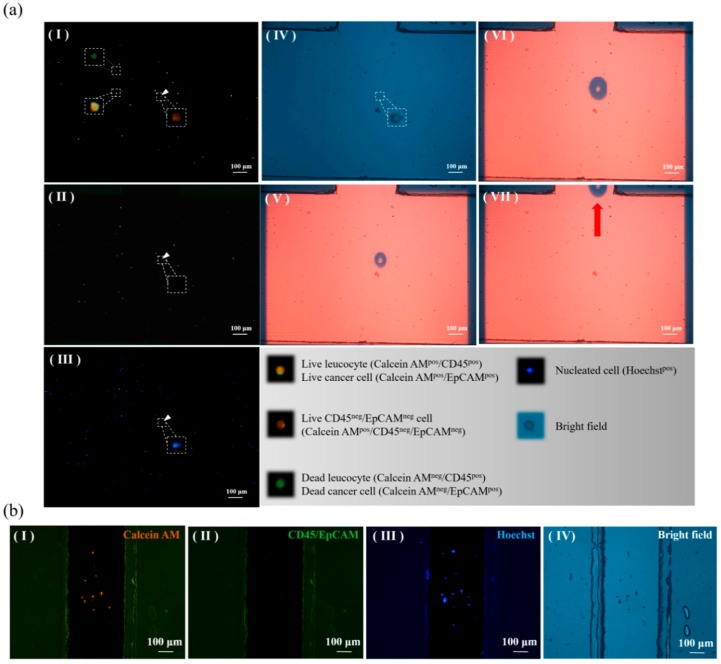
(**a**) Microscopic images showing the isolation of CD45^neg^/EpCAM^neg^ cells from the blood sample of head-and-neck cancer patients: (I)–(III) the processes of the fluorescence microscopic observation to identify the CD45^neg^/EpCAM^neg^ cells ((I)–(II) the orange colour-only dot image and (III) the blue colour dot image), (IV) the target CD45^neg^/EpCAM^neg^ cell was positioned using light field microscopy imaging, (V)–(VII) ODEP-based cell manipulation was carried out to isolate the targeted cells to the side microchannel (a video clip is provided in the Appendix A); (**b**) microscopic observations for evaluating the purity the live CD45^neg^/EpCAM^neg^ cells isolated and collected in the side microchannel ((I) calcein AM-positive live cells (orange colour dot images), (II) CD45 surface marker-positive leucocytes, or EpCAM surface marker-positive CTCs (green colour dot images); (III) Hoechst-positive nucleated cells (blue colour dot images); (IV) the cell image obtained using light field microscopy).

**Table 1 micromachines-09-00563-t001:** Gene expression status of CD45^neg^/EpCAM^neg^ nucleated cells isolated from the blood samples of healthy donors and head-and-neck cancer patients.

Biological Function	Gene Name	Healthy Donor	Head-and-Neck Cancer Patient
		Positive/Total	
EMT-related	EpCAM	0/5	0/8
	CK19	0/5	1/8
	Vimentin	1/5	6/8
	SNAIL1	0/5	1/8
MDR-related	MRP1	0/5	3/8
	MRP2	0/5	0/8
	MRP4	0/5	0/8
	MRP5	0/5	2/8
	MRP7	0/5	0/8
CSC-related	NANOG	1/5	2/8
	OCT4	1/5	4/8

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
