# Peer review of "An Optically Induced Dielectrophoresis (ODEP)-Based Microfluidic System for the Isolation of High-Purity CD45neg/EpCAMneg Cells from the Blood Samples of Cancer Patients—Demonstration and Initial Exploration of the Clinical Significance of These Cells"

_micromachines, 2018, doi:10.3390/mi9110563_

Round 1

Reviewer 1 Report

The authors present a system for isolating subpopulations of cells based on their fluorescence with a focus on CTCs. The authors have optimised the ODEP parameters to allow for rapid isolation of identified cells and used the optimised system for isolating and characterising primary patient samples to investigate the gene expression of CD45 neg EpCAM neg cells in healthy and cancer patient samples.

Comparisons are made with magnetic sorting techniques, however it seems that in this instance FACS would be the system for comparison as gating for EpCAM neg, calcein AM positive, Hoechst positive would be a straightforward gating strategy and should yield high purity and high throughput isolation.

The authors mention that CTCs are very rare cells, yet in their system, they are very abundant - in the methods they should describe the numbers of other cells present and the relative abundance of the target cells for comparison. The state they spike in 2,000 cells, but the numbers of other cells are not stated.

When considering the healthy vs patient data, is there a difference in the abundance of CD45 neg / EpCAM neg cells? How many cells were isolated for each sample? What other cells are CD45 neg / EpCAM neg within a healthy donor sample and is it possible to differentiate these from actual CTCs?

The recovery rate of the device is also not stated – of the cells which are moved into the outlet channel, how many end up in the final gene expression analysis?

Reviewer 2 Report

this is an interesting article in which authors reported isolation of a subset of CTCs with down regulated major proteins. They have reported isolation of stem cells like CTC and confirmed their results using molecular analysis. although their device throughput is not high, I believe this work can be a good read for the community 

Reviewer 3 Report

Liao et al. describe an optically induced (ODEP) method for isolation of CTCs.  The authors use the ODEP method isolating live CD45/EpCAM cells with purity as high as 100% which is much better than existing techniques.  As a demonstration, they looked at cancer related gene expression of these cells and compared the expression profiles between healthy and cancer patients.  They show that there is high expression of vimentin in cancer patients and not in healthy patients.  Overall, this reviewer is very excited for this work and believes there is high potential for this work and only have a few minor questions/comments that can be added into their manuscript.

1.     Is the system fully automated?  Is there an automated link between the fluorescence and the sorting of the cells?  Please state this in their methods/results and discussion

2.     Are there effects on gene expression caused by the light or the activated potential?

3.     How many cells can the authors sort in time t? i.e. what is the throughput of their system?

4.     Are there points of failure in the system such that the light is activated but the cells do not move?  If so, what are the causes of this?
